# Molecular and Functional Roles of MicroRNAs in the Progression of Hepatocellular Carcinoma—A Review

**DOI:** 10.3390/ijms21218362

**Published:** 2020-11-07

**Authors:** Kyoko Oura, Asahiro Morishita, Tsutomu Masaki

**Affiliations:** Department of Gastroenterology and Neurology, Kagawa University, 1750-1 Ikenobe, Miki 761-0793, Japan; kyoko_oura@med.kagawa-u.ac.jp (K.O.); tmasaki@med.kagawa-u.ac.jp (T.M.)

**Keywords:** microRNA, non-coding RNA, hepatocellular carcinoma, biomarker, therapeutic target, therapy resistance, HBV, HCV, fibrosis, inflammation

## Abstract

Liver cancer is the fourth leading cause of cancer deaths globally, of which hepatocellular carcinoma (HCC) is the major subtype. Viral hepatitis B and C infections, alcohol abuse, and metabolic disorders are multiple risk factors for liver cirrhosis and HCC development. Although great therapeutic advances have been made in recent decades, the prognosis for HCC patients remains poor due to late diagnosis, chemotherapy failure, and frequent recurrence. MicroRNAs (miRNAs) are endogenous, non-coding RNAs that regulate various molecular biological phenomena by suppressing the translation of target messenger RNAs (mRNAs). miRNAs, which often become dysregulated in malignancy, control cell proliferation, migration, invasion, and development in HCC by promoting or suppressing tumors. Exploring the detailed mechanisms underlying miRNA-mediated HCC development and progression can likely improve the outcomes of patients with HCC. This review summarizes the molecular and functional roles of miRNAs in the pathogenesis of HCC. Further, it elucidates the utility of miRNAs as novel biomarkers and therapeutic targets.

## 1. Introduction

### 1.1. Hepatocellular Carcinoma

Liver cancer is the fourth leading cause of cancer deaths globally, after lung, colorectal, and stomach cancers [1]. Hepatocellular carcinoma (HCC) is the major subtype, accounting for more than 80% of primary liver cancers [2]. Multiple risk factors are associated with the occurrence of HCC, including chronic viral hepatitis B (HBV) or C (HCV) infections, which account for 80% of HCC cases worldwide [3]. Although alcoholic liver disease (ALD) is a common cause of HCC in the USA and Europe, the numbers of patients with nonalcoholic fatty liver disease (NAFLD) have recently increased in most developed countries, and NAFLD is now frequently listed as a cause of HCC [4,5]. Great therapeutic advances have been made in recent decades; however, the prognosis for HCC patients remains poor with a 5-year survival rate of 15–38% in the USA and Asia, attributed to late diagnosis, chemotherapy failure, and frequent recurrence [6,7,8]. Therefore, exploring the detailed mechanisms underlying HCC development and progression is an effective means of improving the outcomes of HCC patients.

### 1.2. MicroRNAs

Non-coding RNAs lack sequences that encode proteins or peptides; these are classified into small non-coding RNAs of about 20–30 bases and long non-coding RNAs of several hundred kbp. Small non-coding RNAs include microRNAs (miRNAs), small interfering RNA (siRNA), and PIWI-interacting RNA (piRNA). Among these, miRNAs are endogenous, small non-coding RNAs (approximately 19–25 nucleotides in length) that regulate gene expression through degradation of messenger RNAs (mRNAs) or inhibition of translation by binding to the 3′ untranslated region (UTR) of target mRNAs. miRNAs control cell proliferation, migration, invasion, and development in HCC by acting as tumor promoters or suppressors [9,10]. Each miRNA is thought to be capable of post-transcriptionally repressing hundreds of target genes; they are thus strong regulators of gene expression.

The biogenesis of miRNAs, and the mechanisms whereby they regulate translation by binding their target mRNAs, are illustrated in Figure 1. Primary miRNAs (pri-miRNA) are transcribed from DNA in the nucleus by RNA polymerase II and are approximately 3–4 kilobases long with a hairpin structure. They are processed by a nuclear RNase III enzyme (Drosha) and its partner protein DiGeorge syndrome critical region 8 (DGCR8), resulting in intermediate pre-miRNAs of approximately 70 nucleotides in length. Then, exportin-5 and its partner Ran-GTP bind to the pre-miRNA, and the complex is exported from the nucleus to the cytoplasm. The hairpin pre-miRNA is cleaved by ribonuclease III (Dicer) and the transactivation response element RNA-binding protein (TRBP), and is processed into a double-stranded mature miRNA of approximately 22 nucleotides in length. The double-stranded miRNA is then unwound, and one strand forms the RNA-induced silencing complex (RISC) with argonaute (Ago). The miRNA separates into mature, single-stranded miRNA, which is selected depending upon its stability, while less stable miRNA strands are degraded. Complementary base pairing between the seed region of the mature miRNA and the target mRNA results in the degradation of the mRNAs and translational repression [11,12].

miRNAs regulate approximately 30% of human genes via the pathway described above, many of which regulate various carcinogenic molecules and pathways or are located on unstable loci [13,14]. miRNAs have been shown to be dysregulated in most cancers. Oncogenic miRNAs (OncomiRs) and tumor-suppressive miRNAs are associated with carcinogenesis and malignant transformation contributing to or repressing the cancer phenotype. Overexpression of oncomiRs have been observed in various cancers [15]. Dysregulation of miRNAs in cancer has frequently been described in gastrointestinal [16,17], urological [18,19], gynecological [20], and lung cancers [21]. Increasing numbers of recent reports have described the utility of miRNAs as biomarkers or therapeutic targets in HCC [22,23]. In this review, we summarize the molecular and functional roles of miRNAs in the pathogenesis of HCC. Further, we describe the potential for using miRNAs as biomarkers and novel therapeutic targets in HCC, based on recent reports.

## 2. The Role of miRNAs in Liver Regeneration

The liver has an exceptionally high regeneration capacity compared to other human organs. Therefore, in cases of liver damage, hepatocyte loss after hepatectomy, or secondary liver dysfunction, hepatocytes in the remaining liver tissue expand and proliferate to restore liver volume and function. Liver regeneration is a complex and regulated phenomenon in which multiple cell types that make up the liver interact via various signaling factors. Although the mechanisms of liver regeneration in the injured liver are clinically important, much remains unknown. Here, we summarized recent findings on the roles and importance of cell proliferation and apoptosis in liver regeneration.

### 2.1. Cell Proliferation in Hepatocytes

During regeneration and repair following liver injury, hepatocytes undergo division in three stages—initiation, growth, and termination. During initiation, hepatocytes are stimulated by cytokines such as interleukin-6 (IL-6) and tumor necrosis factor-α (TNF-α) shift from the G0 cell-cycle phase to the G1 phase. In the subsequent growth phase, G1-phase hepatocytes proliferate by promoting the cell cycle in a cyclin-dependent manner, under stimulation by the hepatocyte growth factor (HGF). When liver volume and function return to normal after the growth phase, hepatocytes are stimulated by TGF-β and actin to terminate proliferation and return to the G0 phase [24]. Liver regeneration is precisely controlled by various molecular mechanisms. In particular, miRNAs have recently been documented to play key roles in the processes of cell proliferation [25,26,27,28,29].

The role of miRNAs and general miRNA changes during the initiation phase has long been known. One study revealed that approximately 40% of miRNAs had increased 3 h after partial hepatectomy, including those that targeted miRNA synthesis (Drosha, DGCR8, Dicer, and TRBP), whereas approximately 70% had declined at 24 h [25]. A transient increase in miRNAs acts as an initiation signal for cell proliferation, thus providing negative feedback that causes general miRNA changes after 24 h and promotes cell proliferation.

During the growth phase, the abundance of several miRNAs also declines; the levels of most miRNAs were lower within 3 d after partial hepatectomy, and gene expression related to cell cycle and proliferation were accordingly higher [26]. In contrast, several studies have reported increases in miRNA abundance during the growth phase, in which increased miR-21 plays an important role. miR-21 has been shown in vitro to control the transition from G1 to the S phase in primary hepatocyte proliferation and accelerate rapid S-phase entry by targeting a negative cell proliferation regulator, phosphatase, and tensin homolog deleted on the chromosome (PTEN) [27]. Further, an in vivo study demonstrated that miR-21 is a critical regulator of liver regeneration and its upregulation contributed to hepatocyte proliferation by targeting PTEN [28]. It was recently shown that the loss of Dicer1 inhibited liver regeneration by downregulating Dicer1-dependent miRNAs, including miR-21. Further, the introduction of miR-21 restored liver regeneration by inhibiting PTEN and Ras homolog family member B (RhoB). RhoB facilitates activation of the phosphoinositide 3-kinase (PI3K)/AKT/mammalian target of rapamycin (mTOR) axis [29].

Many miRNAs suppress the proliferation of hepatocytes after the initiation phase and induce liver regeneration until the termination phase is reached. miR-23b, which promotes cell proliferation by small mother against decapentaplegic targeting 3 (SMAD3) and inhibiting the TGF-β pathway, was highly expressed up to 24 h after partial hepatectomy; however, its expression subsequently declined consistently from 3 to 7 days [30]. In contrast, miR-34a, which acts as a growth inhibitor, was consistently overexpressed up to 24 h after partial hepatectomy [31]. Blocking or stimulating miRNA pathways during liver regeneration may provide novel therapeutic strategies for managing liver regeneration [32].

### 2.2. Apoptosis in Hepatocytes

Apoptosis refers to highly regulated cell death; it occurs not only in morphogenesis during development, but also serves to control cell numbers and to remove damaged cells in liver steatosis, inflammation, and fibrosis. It, therefore, plays an important role in hepatocarcinogenesis. Several miRNAs regulate programmed cell death via intrinsic (BCL2 and MCL1) and extrinsic (TNF-related apoptosis-inducing ligand; TRAIL and Fas) regulatory pathways and the p53-mediated and endoplasmic reticulum stress-induced apoptosis pathway [33]. The following section describes the primary roles of miRNAs in regulating the key molecules involved in the hepatocyte apoptotic pathway.

miR-15b and miR-16 are TNF-related apoptosis regulators via the antiapoptotic protein BCL2. In severe liver inflammation, miR-15b and miR-16 were upregulated in liver tissues and regulated BCL2 at the protein level; moreover, inhibition of these miRNAs reduced TNF-mediated apoptosis in the liver [34]. Recent evidence indicates that miR-15b can inhibit cell proliferation and downregulate BCL2 mRNA and protein expression in hepatocytes [35].

Abnormal expression of miR-125b is common in various cancers; miR-125b plays dual roles to induce or inhibit apoptosis, depending on the cellular state. Similarly, miR-125b downregulation is frequently observed in HCC development; low expression of miR-125b in HCC tissues correlates with a higher rate of apoptosis [36]. miR-125b is thought to promote apoptosis by directly targeting BCL family members such as MCL1, BCLw, and IL-6R [36]. Therefore, miR125b downregulation may promote malignant transformation and tumor development.

miR-221 and its paralog miRNA-222 have been shown in several studies to aggravate hepatocarcinogenesis by targeting apoptosis-related factors, such as p53, p53 upregulated modulator of apoptosis (PUMA), NF-κB, and signal transducer and activator of transcription 3 (STAT3) [37,38]. During hepatocellular carcinogenesis in vitro and in vivo, miR-221/222 induced TRAIL resistance by regulating tumor suppressors such as PTEN and tissue inhibitor of metalloproteinase 3 (TIMP3), and enhanced cell migration by activating the AKT pathway and metallopeptidase activity [39]. In mice, overexpression of miR-221 led to antiapoptotic effects and delayed liver failure progression by regulating the expression of PUMA in the liver [40].

## 3. The Role of miRNAs in Liver Diseases Associated with HCC

### 3.1. Hepatic Lipid Metabolism

The liver controls the metabolism of fatty acids and triglycerides, and plays an important role in regulating whole-body lipid homeostasis. Excessive accumulation of lipids in the liver is characteristic of several liver diseases, such as ALD and NAFLD, which includes nonalcoholic steatohepatitis (NASH). There is some evidence that miR-122 and miR-34a are important molecules in maintaining metabolic homeostasis in the liver.

miR-122 is highly expressed in the liver specifically, accounting for 70% of its total miRNA content. More than 66,000 copies of miR-122 per hepatocyte are present in the liver, and miR-122 is generally most highly expressed in any liver tissue, including normal and fatty livers [41]. In the liver, miR-122 regulates the expression of various genes related to lipid metabolism [42]. In healthy mice, miR-122 inhibition reduced the expression of hepatic lipogenesis genes, which encode two rate-limiting enzymes, fatty acid synthase (FASN) and acetyl-CoA carboxylase 1 (ACC1). However, miR-122 inhibition in obese mice reduced hepatic lipogenesis and protected against liver steatosis [41]. Furthermore, miR-122 overexpression reduced the expression of insulin-like growth factor (IGF)-1R. Hepatocyte nuclear factor (HNF)-4α is a key regulator of miR-122 activity and of various targets associated with gluconeogenesis and lipogenesis [43]. miR-122 expression was lower in liver samples [44,45] but higher in serum [46,47,48] from patients with NAFLD compared with healthy controls. These results concurred with those of an earlier study, in which miR-122 inhibition caused a significant decrease in serum cholesterol levels and improved hepatic steatosis in high-fat diet mice. This was associated with the activation of adenosine 5′-monophosphate-activated protein kinase (AMPK), increased hepatic fatty acid oxidation, and decreased hepatic lipid synthesis rates [49].

miR-34a targets hepatic sirtuin 1 (SIRT1), which modulates energy metabolism by activating peroxisome proliferator-activated receptor (PPAR) α and liver X receptor (LXR) [44,50]. SIRT1 is also downregulated in NAFLD, and miR-34a inhibition restores SIRT1 expression, which leads to the activation of AMPK [51]. A recent study found that induction of miR-34a, which also regulates caspase-2 levels, contributed to liver steatosis in both humans and mice [52]. Similarly, miR-33, miR-103, miR-104, and miR-307 reportedly modulated lipid and cholesterol regulatory genes [53]. miR-33a and miR-33b, pivotal regulators of lipid metabolism in the liver, play important roles in lipid metabolism with respect to sterol regulatory element-binding protein (SREBP) genes.

### 3.2. Hepatic Inflammation

Liver inflammatory disease is caused by an immune response resulting from various factors, including bacterial and viral infections, alcohol abuse, metabolism disorders, drugs, and diet. Hepatic inflammation and the immune response cause the production of inflammatory mediators, including TNF, IL-1, and IL-6 in macrophages and endothelial cells, which play key roles in inflammatory regulation. In particular, miRNAs such as miR-122, miR-132, and miR-155 have important functions in innate and adaptive immunity associated with liver inflammation. As previously mentioned, miR-122 not only regulates cholesterol and lipid metabolism but also has important roles in hepatic inflammation. miR-122-deficient mice displayed hepatic inflammation and fibrosis, suggesting that miR-122 plays an anti-inflammatory role. This pathology is associated with activation of the carcinogenic pathway, and with hepatic infiltration by inflammatory cells to release protumorigenic mediators such as IL-6 and TNF. Further, miR-122 strongly inhibits tumor formation [54,55]. miR-132 is a well-known mediator in chronic liver disease and has been shown to enhance interaction between adipocytes and inflammatory cells during malnutrition through the inhibition of SIRT1 and deacetylation of p65 [56]. Suggesting the intracellular role in liver inflammation, long-term alcohol administration in mice induced miR-132 throughout the liver and in isolated hepatocytes and Kupffer cells [57]. miR-155 is also expressed in Kupffer cells after alcohol administration, and TNF-induced miR-155 potentially worsens liver inflammation. Elevated serum miR-155 has been found following liver inflammation in individuals with excessive alcohol intake and mice with alcoholic liver injury [58,59].

### 3.3. Hepatic Fibrosis

Hepatic fibrosis progresses with the exposition of the extracellular matrix due to alcohol abuse, cholestasis, and parasite and hepatitis virus infections [60]. Activated hepatic stellate cells (HSCs) in the liver are known to be major effectors associated with hepatic fibrosis. Activation of HSCs via multiple signaling pathways causes severe inflammation and miRNAs are known to modulate various growth-factor receptor signals [61]. miR-21 can mediate activation of human HSCs during hepatic fibrosis via the PTEN/AKT pathway [62]. Patients with chronic hepatitis including *Schistosoma japonicum*-related hepatic fibrosis exhibit downregulation of SMAD7, a negative regulator of transforming growth factor (TGF)-β signaling, which modulates fibrosis and TGF-β, in turn, induces miR-21 [63,64]. Recently, it was reported that miR-96, like miR-21, promotes schistosomiasis-related hepatic fibrosis by activating the SMAD signaling pathway, and increases collagen expression by targeting SMAD7 [65].

miR-221 has been shown to play a key role in liver fibrosis; many reports have demonstrated that it is upregulated in hepatic fibrosis [66], and that it is an effective non-invasive biomarker for evaluation of HCV- or NASH-induced hepatic fibrosis [67,68,69]. In vivo deletion of the miR-221/222 cluster in the liver suppressed TIMP3, a secreted protein that binds to the extracellular matrix and blocks activation of several cell membrane metalloproteases, including a disintegrin and metalloproteinase 17 (ADAM17) and TNF-α converting enzyme; suppression of TIMP3 by deleting the miR-221/222 cluster inhibited hepatic fibrosis [70,71].

Among the other miRNAs associated with hepatic fibrosis, miR-181b promotes hepatic fibrosis via the TGF-β or NF-κB pathways, whereas miR-29b, miR-101, and miR-214-3p suppress hepatic fibrosis by suppressing collagen production in the extracellular matrix or by inhibiting the TGF-β pathway [72]. miR-29a was downregulated in the serum of hepatic fibrosis patients [73]. miR-29a overexpression obstructed signaling of toll-like receptor 2(TLR2) and TLR4, key mediators of hepatic fibrosis in Kupffer cells and HSCs, in liver tissues of cholestatic mice [74].

Various miRNA signatures in liver diseases have been recently identified (Table 1).

### 3.4. Hepatitis B Virus

HBV infection causes a spectrum of liver disease depending on the balance between virus replication ability and host immune status; that is, it progresses from the asymptomatic carrier state, acute hepatitis, chronic hepatitis, and eventually to liver cirrhosis and/or HCC. The host immune response to eliminate HBV elicits cytotoxic T-cell responses with the production of inflammatory cytokines, including free radicals, interferon, and TNF, resulting in liver injury. Integration of HBV into the host hepatocyte genome is important for hepatocarcinogenesis, which triggers apoptosis, regeneration, early senescence, and genomic instability [87].

miRNAs directly target HBV transcripts. miR-199-3p binds directly to the S protein-coding region and miR-201 binds to the pre-S region, resulting in suppression of HBV replication [77]. Several miRNAs target HBV transcripts and regulate HBV-associated liver carcinogenesis by suppressing the wingless-related integration site (WNT)/β-catenin, Janus kinase (JAK)/STAT, and PI3K/mitogen-activated protein kinase (MAPK) pathways [88].

miR-122 targets the mRNA coding for HBV polymerase and the 3′ UTR of the mRNA for the core protein [89]. However, it is well known that miR-122 also targets certain proteins; the downregulation of WNT1 by miR-122 leads to the downregulation of β-catenin [90]. The reduction in miR-122 expression in HBV can upregulate cyclin G1, which attenuates p53 activity, thereby increasing HBV replication [75]. This key miRNA can also downregulate the suppressor of cytokine signaling 3 (SOCS3), which negatively regulates JAK/STAT signaling. Therefore, inhibition of miR-122 induced by chronic HBV infection upregulates SOCS3 and leads to inactivation of interferon, thus promoting viral persistence and hepatocarcinogenesis [91].

miR-155, which controls the generation of class-switch B cells and c-myc-IgH translocation, acts as an immune regulator, controlling cell proliferation, and maintaining homeostasis [92]. A recent study found that increased miR-155 expression levels were positively correlated with measured values of liver enzymes and hepatitis Be (HBe) antigens in the serum of patients with acute HBV infection [93]. HBe antigens increase miR-155 expression in inflammatory cells via the PI3K and NF-κB signaling pathways; this promotes the production of HBe antigen-induced inflammatory cytokines by suppression of BCL6, SH2-containing inositol phosphatase 1 (SHIP1), and SOCS [93].

A recent study revealed that miR-99 family members were highly expressed in HBV-infected liver tissues and that miR-99 levels corresponded with HBV DNA loads in the serum of patients with HBV infection [76]. Transfection of the miR-99 family (miR-99a, miR-99b, and miR-100) promoted HBV replication and antigen expression in HCC cells. Overexpression of the miR-99 family suppressed activation of the IGF-1R/AKT/mTOR pathway and promoted post-transcriptional HBV replication via autophagy through mTOR/Unc-51 like autophagy activating kinase 1 (ULK1) signaling [76].

### 3.5. Hepatitis C Virus

HCV, a hepatotropic RNA virus belonging to the Flaviviridae family, is a primary contributor to liver cirrhosis and HCC with persistent chronic hepatitis. Some miRNAs associated with the host immune response to HCV infection not only regulate the target genes involved in replication, but also directly target the HCV genome [78]. Many host factors, such as poly(rC)-binding protein (PCBP) 2 and miR-122, bind to the 5′ UTR of the viral genome, thus controlling the replication of HCV. Like other miRNAs, miR-122 normally forms a complex with Ago2; it then binds to the 3′ UTR of the target mRNA and suppresses translation or promotes degradation of mRNA, thereby negatively regulating gene expression. In contrast, HCV has two miR-122 binding sites at its genomic RNA 5′ UTR; binding of the miR-122/Ago2 complex to these sites stabilizes the viral RNA and inhibits its degradation [94,95]. HCV infection disrupts miRNA regulation of multiple pathways including the immune system, cell proliferation, signal transduction, and lipid metabolism [96].

Several miRNAs are required for HCV replication. miR-141 significantly promotes HCV replication by downregulating Rho GTPase activating protein (DLC) 1 [79], and miR-491 helps HCV entry through the PI3 kinase/AKT pathway [78]. Further, HCV infection causes changes in the expression of other miRNAs including miR-130a/b, miR-200, miR-34a, miR-23b, miR-24, miR-146a, miR-381, miR-25, miR-200a, and miR-371-5p. The miRNAs regulate the gene expressions associated with PPARγ, STAT3, interferon regulatory factor 1 (IRF1), IGF-1R, fibronectin 1 (FN1), stearoyl-CoA desaturase (SCD), and cAMP-responsive element-binding protein 1(CREB1) [78,97].

Differential miRNA expression has been reported in livers with progressed fibrosis and is involved in HSC activation [98]. Several miRNAs are reportedly useful biomarkers for evaluating the progression of hepatic fibrosis in chronic hepatitis or cirrhosis due to HCV infection. Expression levels of the let-7 family (let-7a-5p, let-7c-5p, and let-7d-5p) were inversely correlated with the degree of hepatic fibrosis due to HCV infection, and low expression of the let-7 family influenced hepatic fibrosis through activation of the TGF-β pathway in HSCs [80]. miR-21 was also a TGF-β-mediated fibrosis modulator in HCV-infected patients with downregulated SMAD7, which suppressed TGF-β signaling [63]. A recent cohort study demonstrated that miR-16, miR-146a, miR-221, and miR-222 were effective biomarkers for predicting the progression of hepatic fibrosis due to HCV. miR-221 and miR-222 are particularly important biomarkers with high sensitivity and specificity in the advanced stage of hepatic fibrosis due to persistent HCV infection [81].

### 3.6. Alcoholic Liver Disease

ALD, including steatosis, alcoholic hepatitis, and liver cirrhosis, is caused by persistent degeneration and necrosis of hepatocytes due to alcohol abuse. The toxicity of alcohol increases the production of reactive oxygen species, resulting in hepatocyte damage and hepatitis due to free radicals and oxidative stress. Since ALD may eventually cause HCC, it is an important condition in the study of hepatocarcinogenesis. ALD often progresses through immune mechanisms including the TLR4-NF-κB pathway. Moreover, TLR4 triggers the activation of MAPK or TIR-domain-containing adapter-inducing interferon-β (TRIF) [99]. Chronic overdose of alcohol intake reduces miR-155 expression and inhibits the expression of multiple TLR4-NF-κB regulators including SHIP1 and SOCS1 [84]. Moreover, several miRNAs (miR-27a, miR-30a, miR-103, miR-107, miR-122, miR-182, and miR-192) are known to regulate inflammatory cytokines associated with ALD [84].

A recent study suggested that let-7 family members, including let-7a and let-7b, were downregulated in mouse livers following ethanol consumption. These miRNAs directly targeted Lin28B and high-mobility group AT-hook 2 (HMGA2). Furthermore, low expression of Lin28B induced overexpression of the let-7 family and suppressed HSC activity and hepatic fibrosis in mice with ALD [82]. In an in vivo study, miR-19b levels were reduced in liver tissues from rat models of alcoholic-induced liver injury, concomitant with a significant increase in pri-miR-17-92 expression and enhancement of profibrotic markers [83]. miR-182 was one of the most overexpressed miRNAs in patients with alcoholic hepatitis compared with patients with other chronic liver conditions; its expression was correlated with the degree of disease progression and short-term mortality [100].

### 3.7. Nonalcoholic Fatty Liver Disease

NAFLD results from fat accumulation in the liver, mainly due to obesity and lifestyle-related diseases in the absence of alcohol abuse, hepatitis viral infection, or other specific causes. It includes the pathology of NASH, which refers to steatosis with inflammation caused by triglyceride accumulation, and can progress to fibrosis, cirrhosis, and hepatocarcinogenesis over a long period. The pathogenesis of NASH, which remains to be fully elucidated, is associated with insulin resistance, dysfunction of adipokine secretion, increased fatty acid β-oxidation in mitochondria, accumulation of misfolded proteins in the endoplasmic reticulum stress, excessive fatty acid consumption, iron overload, production of proinflammatory cytokines in macrophages, small intestinal bacterial overgrowth, and genetic factors [84].

Recent studies indicated that the livers of patients with NAFLD exhibited different miRNA profiles compared to healthy controls. NAFLD liver tissues have upregulated expression of miR-31, miR-33a, miR-34a, miR-144, miR-146b, miR-150, miR-182, miR-183, miR-200a, miR-224, and miR-301a, and downregulated miR-17, miR-122, miR-296, miR-373, miR-375, and miR-378c expression [84,101,102]. In particular, miR-34a and miR-122 play pivotal roles associated with the progression of NAFLD, including NASH and hepatocarcinogenesis, as mentioned in Section 3.1 [42,43,49,51]. miR-155 regulates the function of immune cells and is involved in lipid metabolism [103] and ALD [84]. miR-155 activity is reduced by adipogenic transcription factors CCAAT/enhancer-binding protein (C/EBP)-α, C/EBP-β, PPAR-γ, and LXRα in NAFLD patients; further, miR-155 is implicated in carcinogenesis in NAFLD [84,85,86].

## 4. Dysregulation of miRNAs in HCC

Each miRNA targets multiple mRNAs and regulates the expression of multiple genes in a complex manner; these dysregulations are observed in various cancer types [16,17,18,19,20,21,104]. While some miRNAs are downregulated in HCC, acting as tumor suppressors, upregulated miRNAs also occur in HCC, functioning as oncomiRs. Regions encoding miRNAs involved in dysfunction can harbor genetic alterations such as mutations, amplifications, deletions, or fusions. Transcription of some miRNAs is suppressed by carcinogenesis transcription factors such as Myc, while others are epigenetically regulated by DNA methylation and histone modifications. Additionally, suppression of miRNA processing machinery genes, including Drosha, DGCR8, Dicer, TRBP, and Ago2 has been shown to reduce mature miRNA synthesis, leading to hepatocarcinogenesis and HCC development [104].

In HCC, miRNA dysregulation is involved in all clinical stages, and is evident even in early stages; therefore, miRNA profiles may potentially be used to discriminate HCC patients from healthy controls and those with other liver diseases [105]. Various HCC-specific miRNA signatures have been recently identified (Table 2 and Table 3). In a previous study, we reported that miRNA profiles differed between hepatocytes and HCC cell lines [106], suggesting that miRNA profiles have significant value as biomarkers [107,108]. Insights into the roles of miRNAs in HCC development and progression have made miRNAs attractive biomarkers and therapeutic targets for HCC.

### 4.1. Development of HCC

#### 4.1.1. Oncogenic miRNAs

miR-21, an oncomiR, is frequently overexpressed in various cancer types, such as breast, lung, prostate, stomach, colon, pancreas, prostate, and liver cancers [179,180]. It plays key roles in liver diseases, including viral hepatitis, NAFLD, ALD, liver fibrosis, and HCC [181], and has been shown to induce fibrogenesis in muscles and some organs, such as the heart, lungs, and kidneys [182]. In the liver, it promotes collagen synthesis in the extracellular matrix and fibrosis by activating HSCs, resulting in hepatocarcinogenesis as mentioned in Section 3.3 [62,63,64,65]. Clinical data revealed increased miR-21 expression in both tissues and serum of patients with HCC [181,183], and its expression level was significantly correlated with tumor progression [184]. In an in vitro study, miR-21 overexpression promoted cell migration and invasion of HCC by tumor suppressor targeting kruppel-like factor 5 (KLF5) [110]. Another study demonstrated that HCC cells secreted exosomal miRNA-21, which promoted activation of pyruvate dehydrogenase kinase 1 (PDK1)/AKT signaling in HSCs by directly targeting PTEN; PDK1/AKT signaling activation, and promoted cancer progression by causing HCC cells to secrete angiogenic molecules including vascular endothelial growth factor (VEGF), basic fibroblast growth factor (bFGF), matrix metallopeptidase 2 (MMP2), MMP9, and TGF-β [185]. Further, a positive correlation was discovered between miR-21 and high-mobility group box (HMGB), a prototypical damage-associated molecular pattern (DAMP) protein, in HCC cell lines and tissues: HMGB1-upregulated miR-21 expression was shown to depend on the IL-6/STAT3 signaling axis in human HCC [186].

miR-155 is involved in inflammatory liver diseases such as HBV infection and NASH [58,59,92], resulting in increased tumorigenicity [187]. miR-155 mainly functions as an oncomiR by targeting tumor suppressors including SOCS1, tumor protein p53 inducible nuclear protein 1 (TP53INP1), CDC73, von Hippel-Lindau tumor suppressor (VHL), and mutL homolog 1 (MLH1). Moreover, the upregulation of miR-155 increases cell proliferation by activating the Wnt/β-catenin, AKT, NF-kB, and JAK2/STAT3 signaling pathways. Interestingly, it has been reported that miR-155 plays a tumor-suppressive role and omcomiR under different circumstances such as the time of tumor progression. miR-155 acts as a tumor suppressor in human carcinoma by regulating claudin (CLDN) 1 and SMAD2 [188]. In an in vivo study, miR-155 activated STAT3 signaling by targeting SOCS1, leading to MMP9 production and increased HCC invasion [189].

miR-221 is one of the most highly expressed miRNAs in HCC tissues; its overexpression increases the tumorigenicity of p53-/- myc-expressing liver progenitor cells. Further, miR-221 overexpression stimulates growth of tumorigenic murine hepatic progenitor cells targeting DNA damage-inducible transcript 4 (DDIT4), a modulator of mTOR pathway [190]. miR-221 is also associated with apoptosis by targeting tumor suppressors such as PTEN and TIMP3 by activating the AKT pathway and metallopeptidase expression in HCC [30]. Recently, anticancer effects have been demonstrated by positively regulating PTEN, then inactivating the PI3K/AKT signaling pathway by downregulating miR-221 expression, thereby inducing apoptosis of HCC cells [120].

#### 4.1.2. Tumor Suppressive miRNAs

The let-7 family comprises of multiple tumor-suppressive miRNAs that target the RAS family, whose functions have been elucidated [179,191]. Further, they are antifibrotic miRNAs and are components of central miRNAs involved in transcriptional regulation by PPARγ. These miRNAs control the expression of genes related to hepatic fibrosis, resulting in decreased tumorigenicity [80,192]. Notably, expression of the let-7 family was downregulated in HCV-associated HCC [193]; let-7g was remarkably downregulated in HCC with metastasis relative to HCC without metastasis. Further, higher let-7g expression in HCC tissues compared with that in non-HCC tissues was significantly associated with good prognosis in HCC patients [193].

The miR-15 family comprises miR-15a, miR-15b, miR-16, miR-195, and miR-497, all of which share the same seed sequence and target specific mRNAs that have complementary sequences. In human HCC, the miR-15 family is downregulated, acting as a tumor suppressor, and plays a key role in modulating hepatic inflammation by directly targeting I kappa B kinase (IKK) α and TGFβ activated kinase 1 binding protein 3 (TAB3), upstream regulators of the NF-κB signaling pathway [150]. miR-195 inhibits angiogenesis by directly targeting VEGF, Vav guanine nucleotide exchange factor 2 (VAV2), and cell division cycle 42 (CDC42) in HCC cells [151]. A recent study demonstrated that miR-497 regulated the protein kinase B (PKB) signaling pathway by targeting Rictor in HCC cells and inhibited proliferation, invasion, metastasis, and chemoresistance of hepatoma cells via the Rictor/AKT signaling pathway [168].

miR-29 plays an important role as a tumor-suppressive miRNA in various cancers and has been shown to modulate some oncogenic processes such as epigenetic processes, proliferation, apoptosis, fibrosis, angiogenesis, immune response, and metastasis [194]. In an in vivo HCC study, miR-29 overexpression induced apoptosis and notably inhibited tumorigenicity by directly targeting antiapoptotic proteins, such as BCL2 and MCL1 [195]. In human HCC in vivo, miR-101, which targets MCL1, is downregulated, thereby inducing apoptosis and retarding tumor development [196]. Further, as a tumor suppressor, miR-101 targets RHO-associated protein kinase (ROCK); ROCK is an important downstream effector of RhoA GTPase, which regulates actomyosin bundles and focal adhesions that inhibit the epithelial–mesenchymal transition (EMT) and angiogenesis. ROCK2 is frequently overexpressed in HCC and its stable overexpression confers cell motility and invasiveness in vitro and in vivo [197].

miR-122 plays a pivotal role not only in lipid metabolism and HCV replication but also in hepatocarcinogenesis. miR-122 has been demonstrated as a tumor suppressor in various cancers, including HCC, and directly targets cyclin G1. In mice, miR-122-knockout resulted in steatohepatitis and carcinogenesis, with most genetic changes associated with the regulation of lipid metabolism, inflammation, and fibrosis [54]. miR-122 has been reported to be significantly suppressed in HCC tissue and cell lines and target IGF-1R, ADAM10, and pyruvate kinase M2 (PKM2) [198]. Moreover, it is demonstrated that the restoration of miR-122 in HCC cells reverses the tumorigenic properties and prevents HCC development in mice [198].

### 4.2. miRNAs as Biomarkers for HCC

The prognoses of patients with HCC remain some of the worst among all cancers, and many patients are already at an advanced stage by diagnosis. Tumor markers are widely used to screen for HCC in high-risk patients; these include α-fetoprotein (AFP) and protein induced by vitamin K absence or antagonists-II (PIVKA-II). However, diagnosis of HCC using conventional tumor markers has yielded unsatisfactory results: the sensitivity and specificity for differentiation of HCC were 39–64% and 76–91% for AFP, and 41–77% and 72–98% for PIVKA-II, respectively [199]. Therefore, additional complementary biomarkers are needed, particularly those associated with early HCC.

Numerous studies have supported the use of several miRNAs as diagnostic and predictive biomarkers in HCC. Circulating miRNAs exist stably in peripheral blood [200], and are thus more useful biomarkers for evaluation of HCC. miRNAs released from cancer cells and tissues exist in the exosome-encapsulated form, apoptotic bodies, or are bound to serum proteins or lipids [201]. Notably, because blood samples can be obtained noninvasively and repeatedly, and because miRNAs exhibit tissue specificity, circulating miRNAs in serum or exosomes may be more useful biomarkers than those in tissues. Various miRNA signatures as biomarkers for HCC have been recently identified (Table 4).

The potential utility of circulating miR-122 and let-7 in early-stage HCC diagnosis has been reported, with comparable sensitivity to that of AFP based on serum levels in HBV patients with precancerous nodules and early HCC [202]. Serum miR-101 expression was significantly downregulated in patients with HBV-associated HCC, relative to those with HBV-associated liver cirrhosis and chronic hepatitis, indicating that miR-101 is a potential biomarker for diagnosing and distinguishing HBV-associated HCC [203]. miR223-3p and miR-125b-5p have shown potential as effective biomarkers in HBV-associated HCC [204]. In contrast, miR-122 may be a useful biomarker in HCV-associated HCC; screening of miR-122, miR-885-5p, miR-221, and miR-22 revealed that miR-122 demonstrated high diagnostic accuracy for detection of early HCC in liver cirrhosis patients [205]. We previously reported the efficacy of miR-125a-5p as a diagnostic biomarker for HCV-associated HCC [108]. The expression level of certain miRNAs depends on progress of HCC; low miR-296 expression in HCC patients was associated with large tumor size and advanced clinical stage [159]. Similarly, low miR-137 expression was associated with the presence of lymph node metastasis or vascular invasion in HCC [143].

Several miRNAs have demonstrated potential utility as predictive biomarkers of HCC prognosis. Recent studies have shown that high expression of miR-92a, miR-221, miR-487a, or miR-1468 was associated with poor prognosis in HCC patients [113,126,131,206]. In contrast, downregulation of miR-33a, miR-137, miR-194, or miR-940 led to poor prognosis in HCC [143,149,175,207]. High expression of exosomal miR-32-5p, which targets PTEN, indicated poor prognosis of HCC patients [208]. Lower expression of exosomal miR-638 was associated with poorer overall survival among HCC patients [209]. Further, miR-122, miR-148a, and miR-1246 expression levels were significantly higher in serum exosomes from HCC patients than in those from liver cirrhosis patients and healthy controls [210].

### 4.3. miRNAs as Therapeutic Targets for HCC

Several miRNAs have been shown to regulate multiple targets in HCC-related cascades, and miRNA-based cancer therapy is attractive for the development of more practical strategies. There are currently two miRNA-based strategies for HCC, namely (1) replacement or overexpression of HCC-specific miRNAs using mimics and (2) suppression or downregulation of HCC-specific miRNAs using antagonists. In replacement therapy, the deleted or downregulated miRNAs in HCC are restored; this can suppress the translation of pivotal mRNAs that inhibit proliferation or promote apoptosis in HCC cells [211]. Conversely, specifically overexpressed miRNAs in HCC can be inactivated by antagonists in miRNA suppression therapy [212]. However, when nucleic acids such as miRNAs are directly administered, sufficient gene knockdown effects cannot be obtained due to poor cell membrane permeability and nuclease degradation. Therefore, miRNA oligomers are used to regulate miRNA dysfunctions, including antimiRNA oligonucleotides (AMOs), locked-nucleic-acid antisense oligonucleotides (LNAs), miRNA sponges, miRNA masks, nanoparticles, antagomirs, and multiple-target antimiRNA antisense oligodeoxyribonucleotides (MTg-AMOs) [213,214]. The potential therapeutic applications of miRNA targeting have received increasing attention in HCC research.

The delivery of small RNAs, such as miRNA inhibitors or precursors, can be achieved using adeno-associated virus (AAV) vectors. Remarkably, using AAV vectors for delivery of miR-26a, which is lowly expressed in HCC tissues and highly expressed in normal liver tissues, aided suppression of HCC development; further, miR-26a replacement in HCC cells induced cell-cycle arrest by directly targeting cyclin D2 and E2 [215]. In fact, systemic administration of miR-26a using AVV vectors in an HCC mouse model resulted in dramatic protection against HCC progression by inhibiting cell proliferation and inducing apoptosis without toxicity [215].

LNP-DP1, a cationic lipid nanoparticle formulation, is a useful vehicle to restore dysregulated genes in HCC cells by delivering miR-122, which is often downregulated in HCC cells and tissues. In an in vitro study, the LNP-DP1-mediated transfection of miR-122 downregulated more than 95% of the target genes in HCC cells. In an in vivo study, LNP-DP1-mediated siRNAs and miRNAs were effectively taken up by HCC cells; further, LNP-DP1-mediated delivery of the miR-122 mimic suppressed HCC development by regulating target genes and impairing angiogenesis, without systemic toxicity [216].

### 4.4. miRNAs in HCC Therapy Resistance

Current drug therapies for HCC include the chemotherapeutic drugs cisplatin, doxorubicin, and 5-FU, and the molecular-targeted drugs sorafenib, regorafenib, and lenvatinib [217]. Drug therapy for HCC has made great progress in recent decades, but drug resistance is a major cause of treatment failure. Drug resistance is associated with increases in drug efflux, target switching, cell-cycle checkpoint alteration, increased antiapoptotic signals, and DNA damage repair [218]. Many recent studies have indicated that miRNAs affect HCC drug resistance by influencing genes associated with cell proliferation and apoptosis. Elucidating the mechanism of drug resistance acquisition against HCC via miRNAs will help predict drug susceptibility, develop better therapeutic strategies to overcome drug resistance, and improve treatment outcomes for HCC patients. Various miRNA signatures in HCC therapy resistance have been recently identified (Table 5).

Cisplatin, a first-generation platinum chemotherapeutic drug, is a broad-spectrum anticancer drug that binds to DNA required for the proliferation of cancer cells, causing DNA replication and cancer cell self-destruction. Studies have suggested various relationships between miRNAs and cisplatin. In ovarian cancer cells, miR-214 induced cisplatin resistance by negatively regulating PTEN [219], while miR-33a-5p might induce cisplatin resistance via SOCS3 in HCC cells [220]. Therefore, suppressing miR-33a-5p is one of the potential therapeutic strategies to mediate the cisplatin resistance of HCC cells [221]. The comprehensive review by Jones et al. [222] indicated that multiple oncogenes, including miR-130a and miR-182, are significantly elevated both in HCC tissues and cell lines, and promote cisplatin resistance. Further, changes in the expression of miR-96, miR-130a, miR-182, miR-199a, and miR-340 either increased or reduced sensitivity of HCC to cisplatin [14]. Interestingly, miR-363 was shown to be significantly downregulated in cisplatin-resistant HCC cell lines compared to parental cells, and transfection of miR-363 increased the sensitivity of cisplatin-resistant HCC cells to cisplatin-induced apoptosis by targeting antiapoptotic protein MCL1 [223].

Doxorubicin and 5-FU are classic chemotherapeutic drugs used mainly for transcatheter arterial infusion (TAI). ATP-binding cassette (ABC) transporter protein causes tumor sensitivity and resistance; miR-133a and miR-326 target ABC subfamily C1, increasing the sensitivity of HCC cells to doxorubicin [224]. miR-141 expression was higher in 5-FU-resistant cells than in parental cells [225] due to the downregulation of Kelch-like ECH-associated protein 1 (KEAP1) expression and reactivation of nuclear factor erythroid 2 like 2 (NFE2L2)-dependent antioxidant pathways. Notably, several oncogenic miRNAs, such as miR-200a-3p, miR-183, miR-141, and miR-193a-3p, can promote 5-FU resistance; miR-200a-3p enhanced HCC 5-FU-resistance by suppressing the expression of dual-specificity phosphatase 6 (DUSP6) [225,226,227]. In contrast, some tumor-suppressive miRNAs are associated with reducing 5-FU resistance; miRNA-125b and miRNA-195 increased the sensitivity of 5-FU-resistant HCC cells to 5-FU-induced apoptosis by targeting antiapoptotic proteins BCL2 and hexokinase 2 [14]. Notably, some miRNAs increase toxicity; for instance, miR-503 markedly enhanced 5-FU toxicity in HCC by downregulating eukaryotic translation initiation factor 4E (EIF4E) [228].

Sorafenib is a multikinase inhibitor that suppresses proliferation and angiogenesis by targeting v-raf murine sarcoma viral oncogene homolog B (BRAF), RAF1, VEGFR-2/3, and platelet-derived growth factor receptor β (PDGFRβ). HCC resistance to sorafenib is well known; several oncogenic miRNAs, including miR-93, miR-216a, and miR217, promote sorafenib resistance by targeting p21Cip/Waf1, thereby modulating apoptosis and TGF-β signaling [222]. Overexpression of miR-216a/217 acts as a positive feedback regulator for the PI3K/AKT and TGF-β pathways by targeting PTEN and SMAD7, resulting in hepatocarcinogenesis and HCC recurrence. Additionally, activation of PI3K/AKT and TGF-β signaling contributed to the acquisition of miR-216a/217-induced sorafenib resistance in HCC; conversely, blocking these signals overcame sorafenib resistance and prevented metastases in HCC [229]. In contrast, miR-181 induced sorafenib resistance by suppressing Ras association domain family 1 (RASSF1) [230]. miR-494 increased HCC resistance to sorafenib by modulating PTEN and mTOR signaling [231]. Therefore, concomitant inhibition of miR-181 or miR-494 during sorafenib treatment enhances the sensitivity of HCC to sorafenib, thus indicating that these miRNAs may act as potential therapeutic targets for refractory drug-resistant HCC.

## 5. Conclusions

The biological significance and utility of miRNAs in liver disease, especially in HCC, is a rapidly growing field. Accumulating evidence has demonstrated that many miRNAs play regulatory roles in many biological processes associated with HCC, including oncogenesis, tumor development, cell proliferation, and apoptosis. In this article, we reviewed the molecular and functional roles of miRNAs in the development and progression of HCC. Many studies have investigated aberrant miRNA processing and miRNA expression profiles in HCC, including those of circulating miRNAs, which have contributed to the discovery and clinical adaptation of miRNAs as potential biomarkers for diagnosis (particularly early diagnosis) and as prognostic markers for HCC. Furthermore, investigating their molecular and functional roles is likely to be useful for developing therapeutic targets and understanding resistance to conventional therapies. Although miRNAs do not have a direct antitumor effect on HCC, miRNA-based therapy offers a promising perspective compared to classical gene therapy such as induction of a single gene because miRNA exerts antitumor effects by regulating the expression of multiple genes involved in hepatocarcinogenesis. In addition, miRNAs are generally not immunogenic because they do not encode proteins. A combination of conventional and miRNA-based therapy for HCC may be preferable by offering improved gene transfer efficiency and transgene expression.

Although miRNAs have demonstrated potential as biological targets for HCC treatment in preclinical studies, miRNA-based therapy is not yet suitable for clinical practice. Some important problems need to be resolved before clinical application. The first is that due to tumor cell heterogeneity, miRNAs exert different regulatory effects in different types of tumors, and may even show opposite results in different studies. The second is that in vivo research with miRNAs is still relatively scarce; miRNA regulation in vivo may not always be observed due to the complexity of the in-vivo environment. Therefore, extensive in vivo confirmation of miRNA roles in the progression of HCC and their therapeutic effects is required. Finally, because miRNAs are large molecules, further research is needed to effectively administer and deliver miRNAs into tumor cells in the body. Ongoing global miRNA research will be useful for understanding the current state of clinical miRNA application, and for elucidating the biological characteristics underlying diagnosis, treatment, pathology, and prognosis prediction in HCC.

## Figures and Tables

**Figure 1 ijms-21-08362-f001:**
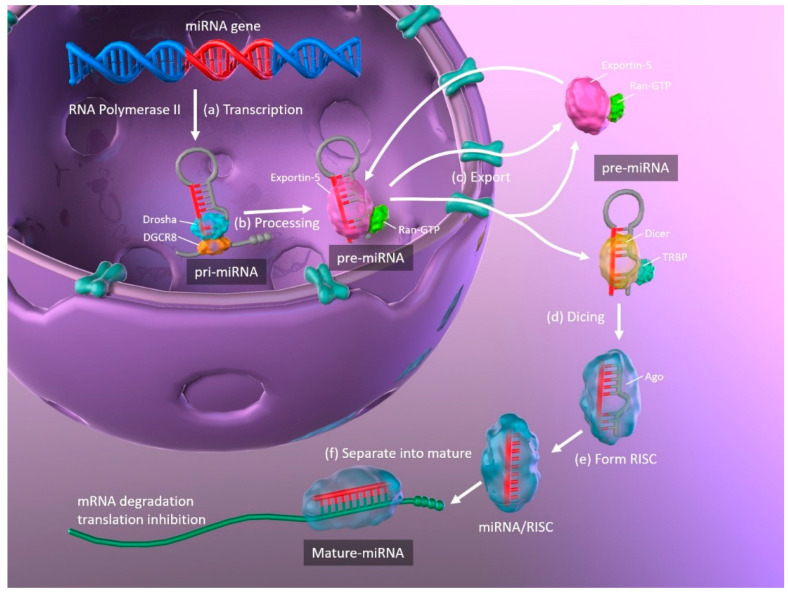
Schematic representation of miRNA biogenesis. (**a**) Synthesis of pri-miRNA transcripts from DNA. (**b**) pri-miRNA is processed by Drosha and DGCR8, resulting in a hairpin intermediate pre-miRNA. (**c**) Exportin-5 and Ran-GTP bind to pre-miRNA, and the complex is exported from the nucleus to the cytoplasm. (**d**) The hairpin pre-miRNA is cleaved by Dicer and TRBP. (**e**) The double-stranded miRNA is unwound and forms RISC with Ago. (**f**) The miRNA separates into mature, single-stranded miRNA. miRNA: microRNA; pri-miRNA: primary microRNA; DGCR8: DiGeorge syndrome critical region 8; TRBP: transactivation response element RNA-binding protein; RISC: RNA-induced silencing complex; Ago: argonaute.

**Table 1 ijms-21-08362-t001:** microRNAs (miRNAs) associated liver diseases.

Liver Disease	miRNA	Function	Reference
HBV infection	miR-122, miR-99 family	Promote HBV replication	[75,76]
	miR-199-3p, miR-201	Suppress HBV replication	[77]
HCV infection	miR-141	Promote HCV replication	[78,79]
	let-7 family, miR-21	Suppress liver fibrosis	[63,80]
	miR-16, miR-146a, miR-221, miR-222	Biomarker for liver fibrosis	[81]
ALD	let-7 family, miR-19b	Suppress liver fibrosis	[82,83]
NAFLD/NASH	miR-155	Promote carcinogenesis	[84,85,86]

**Table 2 ijms-21-08362-t002:** Recently reported upregulated miRNAs acting as oncomiRs in hepatocellular carcinoma (HCC).

miRNA	Targets	Mechanism	Reference
miR-10b	CUB and sushi multiple domains 1 (CSMD1)	Division, migration, invasion	[109]
miR-21	Kruppel like factors 5 (KLF1), Calmodulin regulated spectrin associated protein family member 1 (CAMSAP1), DEAD-box helicase 1 (DDX1), MARCKS like 1 (MARCKSL1)	Division, cell growth, migration, invasion	[110,111]
miR-25	TNF related apoptosis-inducing ligand (TRAIL)	Apoptosis	[112]
miR-92a	F-box and WD repeat domain containing 7 (FBXW7)	Proliferation, cell cycle transition, apoptosis	[113]
miR-96-5p	Caspase-9	Apoptosis	[114]
miR-107	High mobility group AT-hock 2 (HMGA2), 3-hydroxy-3-methylglutaryl-CoA synthase 2 (HMGCS2)	Cell growth, Epithelial-mesenchymal transition (EMT)	[115,116]
miR-135a	Forkhead box O1 (FOX O1)	Migration, invasion	[117]
miR-155-5p	Phosphatase and tensin homolog (PTEN)	Proliferation, invasion, migration, apoptosis	[118]
miR-181a	Autophagy related 5 (ATG5)	Autophagy	[119]
miR-221	P53, P53 upregulated modulator of apoptosis (PUMA), NF-κB, signal transducer and activator of transcription 3 (STAT3), adeno-associated virus serotype 8 (AAV8), PTEN, metalloproteinase inhibitor 3 (TIMP3), TRAIL	Apoptosis, proliferation	[37,38,39,40,120]
miR-203a-3p	Interleukin 24(IL-24)	Cell growth, metastasis	[121]
miR-210	Fibroblast growth factor receptor like 1 (FGFRL1)	Angiogenesis	[122]
miR-214-5p	Wiskott-Aldrich syndrome like (WASL)	Invasion, migration	[123]
miR-302d	Transforming growth factor beta receptor 2 (TGFBR2)	Cell growth, invasion	[124]
miR-346	F-box and leucine rich repeat protein 2 (FBXL2)	Proliferation, migration, invasion	[125]
miR-487a	EVH1 domain containing 2 (SPRED2) 2, phosphoinositide-3-Kinase regulatory 1 (PIK3R1)	Proliferation, metastasis	[126]
miR-765	Inositol polyphosphate-4-phosphatase type II B (INPP4B)	Proliferation	[127]
miR-873	Tumor suppressor in lung cancer 1 (TSLC1)	Proliferation, migration, invasion	[128]
miR-892a	CD266 molecule	Proliferation, invasion	[129]
miR-1249	Patched 1 (PTCH1)	Cell growth, migration, invasion	[130]
miR-1468	Cbp/p300 interacting transactivator with Glu/Asp rich carboxy-terminal domain 2 (CITED2), Up-frameshift protein 1 (UPF1)	Proliferation, apoptosis	[131]
miR-3910	Macrophage stimulating 1 (MST1)	Cell growth, migration	[132]
miR-4417	Tripartite motif containing 35 (TRIM35), pyruvate kinase muscle 2 (PKM2)	Proliferation, apoptosis	[133]

**Table 3 ijms-21-08362-t003:** Recently reported downregulated miRNAs acting as tumor suppressors in HCC.

miRNA	Targets	Mechanism	Reference
miR-15b	B cell CLL/lymphoma 2 (BCL2)	Proliferation, apoptosis	[35]
miR-26	unc-51 like autophagy activating kinase 1 (ULK1)	Autophagy	[134]
miR-29a	Claudin 1 (CLDN1)	Cell growth, migration	[135]
miR-31-5p	Sp1 transcription factor (SP1)	Proliferation, migration, invasion	[136]
miR-33b	Spalt-like transcription factor 4 (SALL4)	Proliferation, metastasis	[137]
miR-98	Enhancer of zeste homolog 2 (EZH2)	Proliferation	[138]
miR-122	A disintegrin and metalloproteinase domain 10 (ADAM10), ADAM17, insulin like growth factor 1 receptor (IGF1R), Serum response factor (SRF), Cyclin G1, Snail family transcriptional repressor 1 (SNAl1), SNAl2	Proliferation, invasion, EMT	[41,139]
miR-125a-5p	Sirtuin 7 (SIRT7), erb-b2 receptor tyrosine kinase 3 (ERBB3)	Proliferation, apoptosis	[140,141]
miR-125b	Myeloid cell leukemia 1 (MCL1), BCLw, IL-6R, SIRT7	Apoptosis, proliferation	[35,141]
miR-126	Vascular endothelial growth factor (VEGF)	Angiogenesis	[142]
miR-137	EZH2	Proliferation	[143]
miR-142	Transforming growth factor β (TGFβ), Thrombospondin 4 (THBS4)	Cell growth, metastasis, migration, invasion	[144,145]
miR-142-3p	Lactate dehydrogenase A (LDHA)	Proliferation	[146]
miR-144	Zinc finger protein X-linked (ZFX)	Proliferation, invasion, migration	[147]
miR-187-3p	S100 calcium binding protein A4 (S100A4)	Metastasis, EMT	[148]
miR-194	Mitogen-activated protein kinase kinase kinase kinase 4 (MAP4K4)	Proliferation	[149]
miR-195	Cyclin D1/3, cyclin-dependent kinase 4/6 (CDK4/6), Cell division cycle 42 (CDC42), Vav guanine nucleotide exchange factor 2 (VAV2), E2F2, BCL2, BCLw, VEGF	Proliferation, apoptosis, angiogenesis, metastasis	[150,151]
miR-199a-3p	VEGFA, VEGFR1-2, Matrix metallopeptidase 2 (MMP2), Hepatocyte growth factor (HGF), Yes 1 associated transcriptional regulator 1 (YAP1)	Angiogenesis, proliferation, apoptosis	[152,153]
miR-199b-5p	TGFβ	EMT	[154]
miR-200a	GRB2 associated binding protein 1 (GAB1)	Invasion, migration	[155]
miR-203	IL1β, SNAl1, Twist family bHLH transcription factor 1 (TWIST1)	Proliferation, metasis	[156]
miR-206	Cyclin D1, CDK6	Proliferation	[157]
miR-212	FOXM1	Migration, cell growth	[158]
miR-296	Fibroblast growth factor receptor 1 (FGFR1)	proliferation, apoptosis	[159]
miR-302b	Epidermal growth factor receptor (EGFR), AKT serine/threonine kinase 2 (AKT2)	proliferation, invasion, metastasis	[160,161]
miR-337	HMGA3	Proliferation, apoptosis	[162]
miR-338-3p	Metastasis associated in colon cancer 1 (MACC1), β-catenin, VEGF	Angiogenesis	[163]
miR-340	Janus kinase 1 (JAK1)	Proliferation, invasion	[164]
miR-345	Interferon regulatory factor 1 (INF1)	Metastasis, EMT	[165]
miR-370	Pim-1 proto-oncogene, serine/threonine kinase (PIM1)	Cell growth, invasion	[166]
miR-451	IL6R	Angiogenesis	[167]
miR-497	RPTOR independent companion of MTOR (RICTOR)	proliferation, migration and invasion	[168]
miR-495	IGF1R	Proliferation, invasion	[169]
miR-539	Fascin actin-bundling protein 1 (FSCN1)	Migration, invasion	[170]
miR-638	SRY-box transcription factor 2 (SOX2), VEGF	Invasion, EMT, angiogenesis	[171,172]
miR-663a	HMGA2	Proliferation, invasion	[173]
miR-874	δ opioid receptor (DOR)	Proliferation, metastasis	[174]
miR-940	C-X-C motif chemokine receptor 2 (CXCR2)	Migration, invasion	[175]
miR-1271-5p	FOXK2	Cell growth	[176]
miR-1299	CDK6	Proliferation	[177]
miR-1301	BCL9, β-catenin, VEGFA	Migration, invasion, angiogenesis	[178]

**Table 4 ijms-21-08362-t004:** Recently reported miRNAs as biomarkers for HCC.

	miRNA	Expression	Background	Reference
Diagnosis	let-7	up	HBV infection	[202]
	miR-101	down	HBV infection	[203]
	miR-122	up	HBV infection, HCV infection	[202,205]
	miR-125a-5p	up	HBV infection, HCV infection	[204]
	miR-223-3p	down	HBV infection	[204]
Poor prognosis	miR-32-5p	up	HBV infection and others	[208]
	miR-33a	down	Unknown	[207]
	miR-92a	up	HBV infection and others	[113]
	miR-122	up	HBV infection, HCV infection	[210]
	miR-137	down	HBV infection and others	[143]
	miR-148a	up	HBV infection, HCV infection	[210]
	miR-194	down	Unknown	[149]
	miR-221	up	HBV infection and others	[206]
	miR-296	down	HBV infection and others	[159]
	miR-487a	up	HBV infection and others	[126]
	miR-638	down	HBV infection, HCV infection and others	[209]
	miR-940	down	HBV infection and others	[175]
	miR-1246	up	HBV infection, HCV infection	[210]
	miR-1468	up	HBV infection and others	[131]

**Table 5 ijms-21-08362-t005:** Recently reported miRNAs in HCC therapy resistance.

Drug	miRNA	Target	Reference
Cisplatin	miR-33a-5p	SOCS3	[220,221]
	miR-363	MCL1	[223]
Doxorubicin	miR-133a	ABC subfamily C1	[224]
	miR-326	ABC subfamily C1	[224]
5-FU	miR-141	Kelch-like ECH-associated protein 1 (KEAP1), Nuclear factor erythroid 2 like 2 (NFE2L2)	[225]
	miR-125b	BCL2	[14]
	miR-195	BCL2	[14]
	miR-503	eukaryotic translation initiation factor 4E (EIF4E)	[228]
Sorafenib	miR-93	p21Cip/Waf1	[222]
	miR-181	Ras association domain family 1 (RASSF1)	[230]
	miR-216a	p21Cip/Waf1	[222]
	miR-217	p21Cip/Waf1	[222]
	miR-494	PTEN, mTOR	[231]

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
