# Peer review of "Molecular and Functional Roles of MicroRNAs in the Progression of Hepatocellular Carcinoma—A Review"

_ijms, 2020, doi:10.3390/ijms21218362_

Round 1
Reviewer 1 Report
- Section 2 (except liver regeneration) also mentioned about liver disease associated with HCC (inflammation and fibrosis). Therefore, it is recommended to re-organized section 2 and 3 (combined section 2 and 3 into one Section in the revised version). In that case, section 2 can be about liver regeneration.
- Authors provided Table 1 and 2. However, it will be possible to summarize miRNAs for liver disease associated with HCC as well. Also, it will be necessary to summarize miRNAs that are involved in resistance and used for biomarkers.
- Authors need to revise ALL abbreviations throughout the manuscript (e.g., DiGeorge syndrome critical region (DGCR) 8, change this style into (DGCR8))
- Line 41: since extracellular RNAs include messenger RNA as well, authors need to revise this sentence.
Author Response
Response to reviewer 1 comments
- Section 2 (except liver regeneration) also mentioned about liver disease associated with HCC (inflammation and fibrosis). Therefore, it is recommended to re-organized section 2 and 3 (combined section 2 and 3 into one Section in the revised version). In that case, section 2 can be about liver regeneration.
Response:Thank you very much for your comments. According to the comment, we have re-organized section 2 and 3.
- Authors provided Table 1 and 2. However, it will be possible to summarize miRNAs for liver disease associated with HCC as well. Also, it will be necessary to summarize miRNAs that are involved in resistance and used for biomarkers.
Response:According to the comment, we have summarized miRNAs associated liver diseases, biomarkers for hepatocellular carcinoma, and drug resistance (revised manuscript Figure1, 4, 5).
- Authors need to revise ALL abbreviations throughout the manuscript (e.g., DiGeorge syndrome critical region (DGCR) 8, change this style into (DGCR8)).
Response:According to the comment, we have revised the abbreviations.
- Line 41: since extracellular RNAs include messenger RNA as well, authors need to revise this sentence.
Response:Non-coding RNAs are classified into small non-coding RNAs of about 20-30 bases and long non-coding RNAs of several hundred kbp. Small non-coding RNAs include microRNAs (miRNAs), small interfering RNA (siRNA) and PIWI-interacting RNA (piRNA). According to the comment, we have revised the relevant sentence.
Reviewer 2 Report
The manuscript provides a good amount of relevant information on the topic; however, the writing style should be significantly improved. There are a number of grammatical errors and poorly organized paragraphs and sentences, so the authors should revise and proofread carefully. Here are my comments:
Throughout the text, when refer to gene names, the authors write full gene names in the sentences followed by abbreviations in the brackets. This creates a lot of broken and unclear sentences and should be revised to improve readability. Usually abbreviations should be in the main sentences followed by full names in the brackets. For example, VEGF (vascular endothelial growth factor).
- Line 18. "is likely to improve" --> "can likely improve"
- Line 44-45: "and in most cases, negatively regulate gene expression in target mRNAs via RNA silencing." This sentence can be removed.
- Line 47: "capable of post-transcriptional repression of" --> "capable of post-transcriptionally repressing"
- Line 52: "that processes pri-miRNAs," should be removed
- Line 59: AGO1 and AGO2 can both form RISC with miRNA
- Line 74-77. The definition of oncomiR and tumor suppressor miRs seems questionable. If not, authors should add a relevant ref to support their definition.
- Line 127: miR-132 is a “chemical mediator”?
- Line 129-131: “Suggesting a cell-specific role….”, this sentence is unclear and should be rephrased.
- Line 132: define “targeting TNF by miR-155”, is TNF a direct target of miR-155?
- Line 136-137: Very awkward sentence with ‘due to’,’results from’,’caused by’. Should be rephrased.
- Line 144-146: “Furthermore, in Schistosoma japonicum-related hepatic fibrosis…” This sentence says almost the same thing compared to the previous sentence.
- Line 166: “multiple cell population types” à “multiple cell types”
- Line 168: “Although the mechanisms of liver-regeneration failure in injured liver are clinically important”, unclear, please revise.
- Line 222: “correlates with the rate of apoptosis”, higher rate or lower rate?
- Line 224: “in anti-cancer therapy” should be removed
- Line 242: “cording” or “coding”?
- Line 245: “disrupting expression” is meaningless, disrupting the expression of key signal transducers?
- Line 275: “miR-122 normally forms a complex with Ago2 and other miRNAs”, miR-122 binds other miRNAs?
- The paragraph in Lines 281-286 is unclear and should be rewritten.
- Lines 303-307: As a reader, here I cannot see any link between TLR4 axis, the several miRNAs mentioned, and ALD.
- Line 347: “advanced” à “evident”
- Contents in paragraph 4.1 are redundant.
- Line 385-388. These two sentences seem contradictory to each other.
- Line 421-423: a broad range of processes were all described as “oncogenic mechanism”, which is inaccurate
- Line 437: if HCV reduces miR-122 (which binds and promotes HCV replication), then why this would promote oncogenicity of HCV? Should it decrease instead?
- Line 485: “transcription”? Should it be translation?
- Line 506: “efferently”? should it be effectively?
- Line 506: “suppression”? should it be delivery?
- Line 561: “several”à “many”
- Line 570: “but regulate expression of multiple genes interacting with other miRNAs”, unclear what this means, please explain.
- Line 583: “efferent”?
Author Response
Response to reviewer 2 comments
- Line 18. "is likely to improve" --> "can likely improve"
- Line 44-45: "and in most cases, negatively regulate gene expression in target mRNAs via RNA silencing." This sentence can be removed.
- Line 47: "capable of post-transcriptional repression of" --> "capable of post-transcriptionally repressing"
- Line 52: "that processes pri-miRNAs," should be removed
Response:Thank you for your comments. According to the comment, we have altered the corresponding part.
- Line 59: AGO1 and AGO2 can both form RISC with miRNA
Response:As point out, RISC is formed by both Ago1 and Ago2. The relevant parts have been changed to “Ago”.
- Line 74-77. The definition of oncomiR and tumor suppressor miRs seems questionable. If not, authors should add a relevant ref to support their definition.
Response: According to the comment, we have changed the sentence to clarify the meaning and also added references.
- Line 127: miR-132 is a “chemical mediator”?
Response:As point out, miR-132 is not a chemical mediator, so we have corrected the word.
- Line 129-131: “Suggesting a cell-specific role….”, this sentence is unclear and should be rephrased.
Response: According to the comment, we have changed the sentence to clarify the meaning.
9.Line 132: define “targeting TNF by miR-155”, is TNF a direct target of miR-155?
Response: miR-155 induction via TNF is correct result of liver inflammation. According to the comment, we have changed the sentence to clarify the meaning.
- Line 136-137: Very awkward sentence with ‘due to’,’results from’,’caused by’. Should be rephrased.
- Line 144-146: “Furthermore, in Schistosoma japonicum-related hepatic fibrosis…” This sentence says almost the same thing compared to the previous sentence.
- Line 166: “multiple cell population types” à “multiple cell types”
- Line 168: “Although the mechanisms of liver-regeneration failure in injured liver are clinically important”, unclear, please revise.
- Line 222: “correlates with the rate of apoptosis”, higher rate or lower rate?
- Line 224: “in anti-cancer therapy” should be removed
Response:According to the comment, we have altered the corresponding part
- Line 242: “cording” or “coding”?
Response:According to the comment, we have corrected it to “coding”.
- Line 245: “disrupting expression” is meaningless, disrupting the expression of key signal transducers?
Response:According to the comment, we have corrected it to “suppressing expression”
- Line 275: “miR-122 normally forms a complex with Ago2 and other miRNAs”, miR-122 binds other miRNAs?
Response:According to the comment, we have corrected the error.
- The paragraph in Lines 281-286 is unclear and should be rewritten.
Response:According to the comment, we have altered the corresponding part to clarify the meaning.
- Lines 303-307: As a reader, here I cannot see any link between TLR4 axis, the several miRNAs mentioned, and ALD.
Response:Chronic overdose of alcohol intake reduces miR-155 expression and inhibits the expression of multiple TLR4-NF-κB regulators including SHIP1 and SOCS1. According to the comment, we have added the sentence to the manuscript.
- Line 347: “advanced” à “evident”
Response:According to the comment, we have corrected it.
- Contents in paragraph 4.1 are redundant.
Response:According to the comment, we have deleted paragraph 4.1.
- Line 385-388. These two sentences seem contradictory to each other.
Response: miR-155 mainly functions as on oncomiR. Although it seems to be inconsistent, it has been reported that miR-155 plays a tumor suppressive role as well as oncomiR. miR-155 plays dual roles in tumor progression as oncomiR or tumor suppressor under different circumstances such as the time of tumor progression. According to the comment, we have altered the corresponding part to clarify the meaning.
- Line 421-423: a broad range of processes were all described as “oncogenic mechanism”, which is inaccurate
Response: According to the comment, we have altered the corresponding part to “oncogenic processes”.
- Line 437: if HCV reduces miR-122 (which binds and promotes HCV replication), then why this would promote oncogenicity of HCV? Should it decrease instead?
Response: The original text was inaccurate. As explained in the HCC-associated liver disease section, miR-122 is strongly related with HBV and HCV replication, but its mechanism in hepatocarcinogenesis and tumor progression remains unclear. According to the comment, we have deleted the relevant inaccuracies and instead added to the revision manuscript an additional role of miR-122 as a tumor suppressor.
- Line 485: “transcription”? Should it be translation?
- Line 506: “efferently”? should it be effectively?
- Line 506: “suppression”? should it be delivery?
- Line 561: “several”à “many”
Response:According to the comment, we have corrected the errors.
- Line 570: “but regulate expression of multiple genes interacting with other miRNAs”, unclear what this means, please explain.
Response:Although miRNAs do not have a direct antitumor effect on HCC, miRNA-based therapy offers a promising perspective compared to classical gene therapy such as induction of a single gene because miRNA exerts antitumor effects by regulating the expression of multiple gene involved in hepatocarcinogenesis.
According to the comment, we have altered the corresponding part as mentioned above.
- Line 583: “efferent”?
Response:According to the comment, we have altered the corresponding part to make it easier to understand.
Round 2
Reviewer 1 Report
The manuscript entitled “Molecular and functional roles of microRNAs in the progression of hepatocellular carcinoma—a review” by Kyoko Oura et al., enhanced the quality of manuscript. The authors have significantly improved the manuscript in an impressively short time.
Author Response
Thank you very much for your comment. we have checked the entire manuscript for possible errors and improved the presentation.
Reviewer 2 Report
The authors addressed most of the points I raised. Still, a number of places need to be further improved, mostly the English language and style (see below). I highly encourage the authors to have someone good at English writing review and proofread their manuscript.
Line 54: "that processes" should be removed
Line 78-79: awkward sentence
Line 209: "inhabitation of SIRT1"? inhibition?
Line 256: "suppressing expression" of what? we do not usually say expression of a pathway, but expression of proteins/genes/molecules
Line 272: what is B6?
Line 294: "have been" --> are
Line 296-298: awkward sentence
Line 411-415: Poorly organized sentences here.
Line 623: efferently should be effectively
Author Response
Thank you very much for your comment. we have checked the entire manuscript for possible errors and improved the presentation with the advice of native speakers.
Line 54: "that processes" should be removed
Response:According to the comment, we have removed the corresponding part.
Line 78-79: awkward sentence
Response:According to the comment, we have altered the corresponding part to clarify the meaning as described below.
“Overexpression of oncomiRs have been observed in various cancers.”
Line 209: "inhabitation of SIRT1"? inhibition?
Response:According to the comment, we have corrected the error.
Line 256: "suppressing expression" of what? we do not usually say expression of a pathway, but expression of proteins/genes/molecules
Response:According to the comment, we have corrected the error.
Line 272: what is B6?
Response:I’m sorry, it was a mistake of “BCL6”. According to the comment, we have corrected the error.
Line 294: "have been" --> are
Response: According to the comment, we have corrected the error.
Line 296-298: awkward sentence
Response: According to the comment, we have altered the corresponding part to clarify the meaning as described below.
“Further, HCV infection causes changes in the expression of other miRNAs including miR-130a/b, miR-200, miR-34a, miR-23b, miR-24, miR-146a, miR-381, miR-25, miR-200a, and miR-371-5p. The miRNAs regulate the gene expressions associated with PPARγ, STAT3, interferon regulatory factor 1 (IRF1), IGF-1R, fibronectin 1 (FN1), stearoyl-CoA desaturase (SCD), and cAMP-responsive element-binding protein 1(CREB1).”
Line 411-415: Poorly organized sentences here.
Response: According to the comment, we have altered the corresponding part to clarify the meaning as described below.
“miR-221 is one of the most highly expressed miRNAs in HCC tissues; its overexpression increases the tumorigenicity of p53-/- myc-expressing liver progenitor cells. Further, miR-221 overexpression stimulates growth of tumorigenic murine hepatic progenitor cells targeting DNA damage-inducible transcript 4 (DDIT4), a modulator of mTOR pathway [190]. miR-221 is also associated with apoptosis by targeting tumor suppressors such as PTEN and TIMP3 by activating the AKT pathway and metallopeptidase expression in HCC [30]. Recently, anticancer effects have been demonstrated by positively regulating PTEN, then inactivating the PI3K/AKT signaling pathway by downregulating miR-221 expression, thereby inducing apoptosis of HCC cells.”
Line 623: efferently should be effectively
Response: According to the comment, we have corrected the error.